# Recommendations for Transdisciplinary Professional Competencies and Ethics for Animal-Assisted Therapies and Interventions

**DOI:** 10.3390/vetsci8120303

**Published:** 2021-12-02

**Authors:** Melissa Trevathan-Minnis, Amy Johnson, Ann R. Howie

**Affiliations:** 1Psychology and Counseling Program, Goddard College, 123 Pitkin Rd, Plainfield, VT 05667, USA; 2Human-Animal Interaction Program, University of Chester, Chester CH1 4BJ, UK; dramyjohnsonlpc@gmail.com (A.J.); humananimalsolutions@comcast.net (A.R.H.)

**Keywords:** human–animal interaction, competencies, ethics, animal welfare, human–animal studies

## Abstract

AAI is a transdisciplinary field that has grown exponentially in recent decades. This growth has not always been synergistic across fields, creating a need for more consistent language and standards, a call for which many professionals in the field have made. Under the umbrella of human–animal interactions (HAI) is animal-assisted interventions (AAIs), which have a more goal-directed intention with animals who have been assessed for therapeutic, educational, or vocational work. The current article offers a brief history and efficacy of HAI, describes the limitations and gaps within the field and recommends a new set of competencies and guidelines that seek to create some of the needed common language and standards for AAI work to address these limitations.

## 1. Introduction

The importance of competencies in specialized areas in addition to general professional competencies is noted in multiple fields. Psychology, for example, documents these standards and competencies in the American Psychological Association (APA) Ethical Principles and Code of Conduct (https://www.apa.org/ethics/code (accessed on 4 October 2021). This article uses the APA code of conduct as a foundation for the material herein and refers to psychologists and mental health professionals as these are the backgrounds of the authors, but the utility of these suggestions to all professionals that partner with animals is emphasized. Within psychology specifically, the establishment of specialty area competencies influences professional training programs and psychologists’ provisions of services. Additionally, such professional competencies affect the public’s perception of the field of psychology. As with other specialty areas in psychology, competencies in animal-assisted interventions (AAI) and animal-assisted therapy (AAT) provide a foundation of credibility for practitioners incorporating AAI strategies in their psychological practice or practice involving animals. The documents herein apply to both AAI and AAT but also any activity or work involving animals. A broad range of animals are considered in this context are, as many species of animals are involved in AAI and AAT across fields and humane treatment and consideration is abundantly necessary for all.

AAIs have evolved and become more reputable over the past 50 years [1,2] Research has established AAIs as an area of evidence-based clinical practice for psychologists and other mental health providers [1]. Therefore, it has become important to identify and document the areas of knowledge, skills, and attitudes needed for a practitioner to be considered competent in AAI and to comply with the APA ethical standards and state licensure regulations.

Other professional organizations have published AAI competencies, such as but not limited to the American Counseling Association, Animal Assisted Intervention International, and International Society for Animal Assisted Therapy. However, until the APA-HAI Competencies document described here [3], psychology has had no delineation of competency in AAI clinical work. This work is also relevant for other fields whereby developing a standard language and set of competencies creates greater ability to collaborate across professions [4]. Additionally, without identified competencies, there is an increased potential for exploitation and increased risk management concerns for clients and practitioners. As stated in the document published by APA Division 17, “without a set of guidelines, the public and professionals will make up their own or practice without guidelines” [3]. The competencies published by APA Division 17 follow the established framework indicated by the APA Code of Conduct. The identified recommended competencies provide needed guidance for training programs and practitioners. It is important that a clear set of guidelines and competencies be considered by any person working within a setting involving AAIs, including but not limited to practitioners and mental health providers partnering with animals, animal handlers, animal owners, AAI assistants, and facilitators. As the field continues to grow and new research emerges, we recommend that these guidelines continue to be evaluated and updated as necessary.

### 1.1. Brief History of AAI

Human–animal studies (HASs) and animal-assisted interventions (AAIs) have experienced exponential growth, especially in recent decades [1,2]. HAS is a transdisciplinary, largely academic field best defined by focusing on and researching relationships with animals, with a smaller subset of social service professionals who incorporate animals into their clinical work with human clients, including but not limited to counselors, social workers, practitioners and those working in vet sciences. This is a notable departure from other social science studies where animals have been used typically as a mechanism for better understanding human psychology, often with little interest in the relationship between humans and animals. While the early Greeks prescribed horseback riding to elicit a positive mood in patients [5] and Sigmund Freud was known for including his dog Jofi in sessions with clients in the 1930s [6], it was not until the 1960s that animal-assisted interventions (AAIs) became the subject of closer observation and investigation. Boris Levenson’s 1969 book *Pet-Oriented Child Psychotherapy*, published despite criticism from colleagues at the time [6], would serve as a catalyst for research suggesting the positive impact of animals on humans [7]. Levinson would go on to be known within the field as the father of animal-assisted therapy. Mental health providers have continued to integrate animals into their practices in a variety of settings and with a variety of populations.

In response to a growing concern for the rights and welfare of animals, Practitioners for the Ethical Treatment of Animals (PSYeta) formed within the American Psychological Association (APA) in the early 1980s, with the goal to protect any perceived threats to animal research including inhumane treatment [8]. Several efforts were made to form a human–animal research division within the APA, but this was met with contentious resistance from a number of groups not in support of animal welfare within research. However, Division 17, sympathetic to the cause, invited petitioners to form a section within their Division in 2005, where the HAI Section 13 is still housed today [8].

### 1.2. Efficacy and Outcomes of AAI

Not surprisingly, the research demonstrates many positive outcomes of incorporating animals into our lives and into our clinical practices [9,10,11]. Previous studies have suggested that AAI work can contribute to reductions in physiological anxiety, depression, and social isolation and an increase in therapeutic participation, and that this can subsequently increase comfort with a facilitator as well as fidelity [2,12,13,14]. Participating in animal-assisted therapy (AAT) can also increase self-esteem, promote social skills development, increase feelings of empathy [15], decrease blood pressure, increase motor functioning [14], and provide an opportunity for safe touch and repair work that can be transferred to other relationships [2]. While relational work with animals is not a good fit for all people or clients, studies have shown that, overall, “a therapeutic relationship with animals is beneficial to psychological well-being” [16] and that clients are more likely to engage in therapeutic interactions when an animal is present [17]. Some authors suggest that the flexible nature of AAIs offers utility for a variety of issues facing clients, from trauma to autism [2] and substance abuse [18], among others. For patients waiting to undergo oncological medical treatment, AAIs are perceived to benefit anxiety, pain, communication, and cognition [19]. Therapeutic growth for children and adolescents in the areas of social, behavioral and emotional issues has been shown when their treatment includes equines [20].

## 2. Increasing Diversity, Animal Welfare, and Professional Competence and Guidelines

The fields in which we find people working alongside animals is extensive, as are the professions in which people work with animals and humans toward clinical goals. In fact, most providers, 91.7% in one study, had heard of animal-assisted therapy and considered it to be a legitimate therapeutic modality [21]. However, despite its popularity, and perhaps in part because of it, terminology, language, and standards vary widely. The development of clear guidelines and standards within AAIs has not kept pace with the growth of the field. This has led to inconsistencies and a lack of clear guidance as mental health providers embark on this work. This in turn creates greater risk to the clients, providers, and animals involved [22]. Fine, Tedeschi and Elvove (2015) [3] point to the need to tighten our identity and bring greater clarity and definition to competencies and training. As noted earlier, Fine, Beck, and Ng (2019) [23] suggest that AAIs are on the brink of a paradigm shift, and a 2021 study came to similar conclusions whereby the field long-regarded as useful but lacking rigor is moving toward greater empiricism and professionalization [24].

A needed shift toward animal welfare is also a notable finding in the literature, as well as a need for more focus on advocacy, diversity, and appreciation of the role of culture and background [24]. Related to multiculturalism within the AAI literature and research, Kaufman in 1999 [25] and later Risley-Curtiss, Holley and Wolf in 2006 [26] point to the paucity of focus on diversity. However, the subfield of AAIs has put much work and effort into empowering marginalized groups via AAIs in populations and settings such as prisons, at-risk youth, and with aging adults among others [16,27,28]. While the AAI concept is ever evolving, there is more work to carry out to make the field accessible and diverse to professionals. In response to this need, there has been a concerted effort to increase awareness and focus on culture and diversity within the field of AAIs, perhaps especially so in recent years. The HAI Section 13 of Division 17 has specifically increased their focus in this area by developing the Inclusion, Diversity, Equity and Access (IDEA) committee which have, in partnership with the social media committee, increased posts related to cultural diversity and HAI practice and research.

Finally, a call for greater consistency across guidelines, ethics, and training has been noted in the research [23,24,29]. Currently, no professional bodies currently ensure that professionals who work with their animals are operating with competence. The Animal-Assisted Therapy in Counseling Competencies document [30], endorsed by the American Counseling Association, was a step toward meeting this need, and the recommended competencies document described in the current article is a further step toward disseminating a more comprehensive language and standards for the field. The hope is that greater consistency across disciplines will create more cohesion and collaboration, but more importantly, a greater understanding of how to engage in AAI work in a way that best protects the welfare and interest of the clients and animals with whom clinicians engage and partner for clinical endeavors [24,31]. Further, the competencies described herein offer explicit focus on advocacy, both for clients but particularly for the nonhuman animals involved in the work [23]. This model of advocacy and focus on welfare is judged by the authors to be a useful model for clients who are often learning how to self-advocate. When we model respect and care and partnership with our animal companions, we, by extension, model that for our clients as well.

One framework that is increasingly popular within the literature and that aligns with our proposed competencies is the application of the One Welfare model to AAIs [32]. The One Welfare framework is underpinned by the notion that humans, animals, and the environment are interconnected [33]. As an extension of the Center for Disease Control’s One Health concept (2016) [32], both ascribe to the notion that humans will not thrive if the welfare of other living animals or the environment is disregarded [33]. In the context of AAIs, the animals as well as the humans should be allowed to consent to the interaction in order to maintain the positive welfare of both and the provider involved in the intervention should have knowledge and education to understand the behavior of their animal at the species, breed, and individual levels [30]. Hediger, Meisser, and Zinsstag (2019) [34] assert that “There is no tradeoff of human benefits against animal health and wellbeing and under which circumstances animals could actually benefit from such interaction with humans” (p. 2). The animals working in an AAI context should gain something from the interactions, otherwise such interactions lead to exploitation [35] and with the animal having the compromised power differential in the relationship with the human, the message sent to the client may be negative or even harmful to the therapeutic relationship.

This emphasis on the welfare and consent of animal participation builds upon the five freedoms of animals refined by the Farm Animal Welfare Committee in 2019 [36] but further suggests the importance of animal enjoyment and consent in any work in which they are asked to engage. Therefore, the competencies described below seek to meet many of the existing gaps and shortcomings of a growing and increasingly efficacious field and directly address the wellbeing of the animals involved. Practitioner or mental health provider (*provider* for short) are used in this article to describe AAI practitioners, clinicians, and therapists.

## 3. Summary of Considerations for APA Ethical Standards Competencies in Animal-Assisted Interventions

The recommended competencies and ethics for the APA-HAI (see Appendix A) were developed by a national committee of 14 highly qualified, expert-level animal-assisted intervention providers with expertise in fields such as psychology, animal behavior, ethology, dog training, clinical social work, education, allied health, and occupational therapy and shared with external AAI experts for review. The team consisted of a working group of The American Psychological Association Human–Animal Interactions Division 17, Section 13 (APA-HAI) Board, which is an internal organization that proffers scholarly and professional activities to advance the practice of human–animal interaction in psychology. While the capacities within which animals can be found in the field of psychology vary widely (e.g., service animals, animals in experimental settings), the APA-HAI division and this document focus exclusively on mental and physical health providers who include animals in their professional practices.

Additionally, these guidelines were built upon existing empirical, foundational, and seminal research [30,37]. These guidelines are recommendations for professional conduct for practitioners (APA, 2010a) and not intended as standards for which professionals must adhere. As within the APA Code of Ethics, the onus is on the provider to have the knowledge, skills, and attitudes necessary to provide competent therapeutic interventions. Additionally, guidelines are intended to strengthen the development of the provider (specialty guidelines for forensic psychology) [38] and rely on the judgment of each individual to reduce the risks associated with any specialty area. Competence in a specialty area requires education, supervised experience, and professional practice (specialty guidelines for forensic psychology) [38].

It is important to point out that there are no accrediting agencies to ensure that professionals who work with animals are operating competently and ethically. The APA-HAI has worked tirelessly to not only develop and disseminate research, evidence-based practices, and educational activities, but also to work alongside national and international human–animal interaction organizations to share consistent messaging, uniform terminology, and empirical evidence to professionals who wish to add an appropriate animal to their clinical settings.

While there are organizations that evaluate and register animals, most commonly dogs and equines, to be a part of an animal-handler team, many of these are volunteer organizations and do not evaluate an animal’s competence in terms of professional practice. In fact, few organizations use competency-based evaluations at all [21]. Further, the competencies an animal needs to work in a volunteer role can be quite different from the competencies needed to work with a professional. Competence—with both humans and animals—is not one-size-fits-all. As a result, it is incumbent upon the professional to determine if the animal evaluation is generalizable to the professional’s setting [30].

Beyond meeting the minimum requirements for licensure in one’s field, the codes of ethics in all health and human service professions include the need for additional education to practice in a specialized area or incorporate a new theoretical framework. For example, if a provider seeks to add eye movement desensitization and reprocessing (EMDR) to their repertoire of services, a client would want to know if that provider has received additional training specific to how EMDR works, how to use the equipment, and that they have been evaluated on their ability to provide a safe, efficacious experience to clients. In AAIs, there is the assumption that the provider has previously achieved proficiency as a practitioner in their work with humans prior to bringing animals into the milieu. Standard 2.01 of the APA Code of Ethics states that where standards do not exist, the psychologist must take “reasonable steps” toward receiving the additional education, experience, and supervision (APA, 2010a) and particularly animal behavior at the species, breed, and individual levels [30,37,39]. AAIs require a specialized skill set that is not offered in traditional academic education for human services.

Providers who want to integrate a live, sentient being into the therapeutic milieu should be subjected to similar criteria; however, without an overarching accrediting body and because there is often a misconception that having a pet or having owned pets makes an individual proficient in animal-assisted interventions, this is often not the case. Bringing in animals who have not been properly selected, prepared, and evaluated comes with many risks and liabilities to the provider.

Finally, within the AAI setting, animals and humans do not share privilege nor power and are subsequently more vulnerable. The Do-No-Harm tenet for human service providers must extend to the animals as well. The welfare of the animal is directly connected to the welfare of the client; thus, protecting the welfare of the animals can send negative messages or intentions to the clients, thereby stifling the therapeutic process rather than helping it. Within the context of AAIs, animals are no longer subjects, but instead are partners in the process who bring with them their own likes, dislikes and limitations [40].

### Development of the Competencies Document and Alignment with Existing APA Code of Ethics

Building upon the American Psychological Association’s Code of Ethics 2.0 Boundary of Competence, the APA-HAI has developed a set of competencies and ethics to help guide providers who want to include animals in practice to be ethical and safe providers of care. Having a set of competencies and ethics not only helps ensure an ethical and culturally proficient practice and protects providers from potential risk but can also help maintain the safety and wellbeing of clients and the animals.

Without a specialty-area-specific set of guidelines, professionals and individuals tend to make up their own. The inclusion and endorsement of professional competencies through the APA-HAI allows providers to make more informed decisions about their abilities and awareness of constructs such as theories associated with AAIs, evidence-based techniques, continual review of the literature, use of uniform terminology, and knowledge of animal behavior specific to the species, breed, and individual levels [30] at a minimum. The overall goal for these guidelines is not to use them as a credentialing tool or checklist for educational curricula; in fact, the breadth and depth of the knowledge, skills, and attitudes aligned with AAIs would be hard to meet within any one program, but instead are part of a continual effort to become a more proficient practitioner. Additionally, the practitioner should acknowledge that AAIs are not a stand-alone intervention, but instead enhance the treatment process. The animal may be the impetus in the client achieving their goal; however, the mechanism of change comes from the provider’s skillset, training, education, and clinical expertise in partnership with the therapy animal.

These guidelines follow the General Principles of APA that include: Principle A: Beneficence and Nonmaleficence, Principle B: Fidelity and Responsibility, Principle C: Integrity, Principle D: Justice and Principle E: Respect for People’s Rights and Dignity and the assumption is that the provider necessitates that these principles are followed. More specifically, the APA Ethical Standard 2.01 Boundaries of competence laid the foundation from which the APA-HAI Competencies were developed using the six clusters: professional (values and attitudes; cultural considerations), relational (interdisciplinary teams), science (related to AAIs and the human–animal bond), application (interventions and consultation), education (teaching and supervision), and systems (leadership, consultation, interprofessional teams) and are illustrated in Figure 1.

## 4. Application of Competencies

Competencies include knowledge, skills, and attitudes—and their integration—allowing an individual to perform tasks and roles as a provider, regardless of the service delivery model [41]. The APA refers to standards as any criteria, protocols, or specifications for conduct, performance, services, or products in psychology or related areas. Competencies are distinctive elements necessary for competence, they correlate with performance, and can be evaluated against agreed upon standards [42]. Standards are considered to be mandatory and may be accompanied by an enforcement mechanism.

The recommended APA-HAI competencies are not to be considered standards, rather they are recommendations intended to guide those interested in developing and implementing AAIs into their practice and assisting providers in making informed choices on how to do so ethically. One might think of these guidelines as a recipe; a recipe does not outline the rules of the kitchen nor does it provide a one-size-fits-all procedure in all cases. Instead, it can be used as a guide map to enable one to successfully obtain a finished product. While one might be able to prepare a dish without use of a recipe, following one will increase the likelihood of a successful finished product. Importantly, a cook might need to tailor a recipe to fit their specific needs and situations. The same is true when conducting AAIs; a particular guideline might have variations in its application depending on the milieu or circumstances at hand. As no professional bodies currently ensure that professionals who work with their animals are operating with competence, criteria that providers may follow in order to achieve best practice are offered here. It is recommended that AAI practitioners follow the guidelines provided in order to minimize risks and liabilities to the provider as well as risk to the animal and client. In order to achieve the goal of minimizing risk, a brief overview of how to prevent misusing or misinterpreting these guidelines is offered below.

### 4.1. Guidelines Are Incomprehensive

Composing a set of guidelines that encompasses every possible consideration of AAIs related to every setting is impossible. Several different types of AAIs exist and can occur in a number of different settings (e.g., schools, hospitals, residential treatment centers, etc.). AAIs may be group or individual in nature and may be implemented with persons of any age. Moreover, research on AAIs, as well as related topics such as anthrozoology and human–animal bond, is ever-growing. Due to this, a practitioner should not assume that this set of guidelines contains all one needs to know or consider for ethical and successful practice and implementation. While it is certainly a good starting point, the provider should consult other materials and experts in instances that a topic or concept is not covered. It is important to seek out guidelines published by other international organizations, such as the International Association for Human–Animal Interaction Organizations (IAHAIO) and Animal-Assisted Intervention International (AAII).

### 4.2. Clinical Judgement Is Key

It is the responsibility of the practitioner to obtain and maintain the knowledge, skills, and attitudes necessary to provide competent AAIs. These guidelines do not provide information on therapeutic or other types of mental health interventions. AAIs are an enhancement to the treatment process rather than stand-alone interventions and can be used by a wide range of professionals (e.g., practitioners, social workers, counselors, etc.). Thus, before one decides to include AAIs in their practice, they should be well-versed and well-practiced in providing the type of intervention at hand without an animal present. The American Psychological Association (APA; n.d.) defines clinical judgement as drawing conclusions from expert knowledge on the appropriateness of treatments and likelihood of improvement. It is well-known by those who work with clients that anything can happen. Oftentimes, clients may do or say things that are unexpected to the provider. When an animal is included in the dynamic, the potential for unexpected interactions increases. It is expected that things may come up in AAI that are not covered by the guidelines. Thus, even with a set of guidelines, it is imperative that individuals rely on their clinical judgement when needed, which they may draw from their previous experience and expertise, to reduce risk.

### 4.3. Continuing Education

As discussed, these guidelines are intended to both inform and strengthen the development of the provider. However, the understanding of and even adherence to these guidelines will not necessarily ensure competence in AAIs. Competence in any area requires continued education (CE), supervised experience, and professional practice; however, not all learning opportunities are equal. The industry standard for CE has moved toward a competency-based model whereby learning outcomes are preferred over instructional input [43], which the APA-HAI Competencies document helps to provide for AAI professionals. This shift ultimately increases the quality of CE programs and allows for assessment-based learning where learning activities are based first on the identified competencies [44]. The professional, in a way, plays the role of a student throughout their career. It is important that anyone facilitating AAIs continues to seek knowledge and consultation from professionals and groups of professionals in the field as well and continue to be informed by recent research, and to seek professional guidance should a question regarding best practice ever arise. Further, it is important to remember that proficiency in one area does not automatically constitute proficiency in another; even if a provider becomes comfortable in one setting or with one animal (or breed or species of animal), that provider should not assume proficiency will remain if they should change practice setting, type of work, or therapy animals.

### 4.4. Rely on Others

The breadth and depth of the knowledge, skills, and attitudes aligned with AAIs cannot be met with any one program and are not encompassed by any one profession. Therefore, knowledge related to issues about a variety of disciplines and the ability to interact with multiple disciplines is required for ethical and effective AAI integration. Providers should be open to considering theories, ideas, and input across disciplines, which might involve consultation with varying organizations.

## 5. To Whom These Competencies Apply

It is important that the competencies be considered by any person working within a setting involving AAIs. This includes, but is not limited to, animal handlers, animal owners, AAI assistants, and facilitators. Consideration of the guidelines can benefit more than just the AAI provider; by reducing risks associated with AAIs, the client and animal will also benefit.

As previously noted, the competencies build upon the APA Standard 2.01 Competence. Within this standard, six clusters are present, one of which is Systems. The Systems domain of competencies encompasses awareness of how varying disciplines interact and confer on issues related to meeting the therapeutic goals of an individual being served across disciplines. These types of interactions are an indicator of how these competencies are important to multiple persons.

Practitioners possess their own professional identity and adhere to their own professional code of ethics. However, AAI-related work often requires the shared collaboration and expertise of professionals across a range of disciplines, such as animal behavior, animal training, veterinary medicine, and other mental health disciplines such as social work. Individuals not only belong to various disciplines but also take on varying roles in supporting the AAIs. Importantly, the animal is also considered a stakeholder in the system and is given priority of care and consideration, as it has the least power in the system. The practitioner is required to advocate for clients and participating in AAIs means advocating for the therapy animal as well. Ethically, practitioners are obligated to advocate for the physical and psychological well-being of the animal. Doing so will benefit all stakeholders involved.

Systems competencies are evidenced through knowledge, skills, and attitudes related to AAIs. Knowledge refers to familiarity with the roles and responsibilities among disciplines involved. Knowledge also refers to familiarity with resources for ongoing education about animal species participants. Skills refers to collaborating with and respecting members of other professions. Here, all stakeholders should be solicited for input in programming. Last, attitudes refer to consideration and respect for all stakeholders involved. Consideration should be given to theories, ideas, and input across disciplines. This might involve consultation with varying organizations. As described, practitioners should also possess knowledge, skills, and attitudes related to welfare of the animal.

## 6. Clinical Risks and Considerations

Practitioners and mental health professionals who are considering providing AAIs must be aware of the inherent risks to the provider, client, animal, public, and the credibility of the field of psychology. The animal’s owner/handler is responsible for liability and other expenses and considerations such as state, local, and federal laws and policies specific to vaccinations, husbandry, licensing, and registrations, regardless of financial or other access barriers. Specific potential risks to animals, clients, the public, and practitioners are identified below.

### 6.1. Ameliorating Potential Risks to the Client

The APA Code of Ethics Principle A: Beneficence and Nonmaleficence states that “Psychologists strive to benefit those with whom they work and take care to do no harm. In their professional actions, psychologists seek to safeguard the welfare and rights of those with whom they interact professionally and other affected persons, and the welfare of animal subjects of research”. The authors wish to emphasize that animals who participate in AAIs are not research subjects. Instead, they are considered partners in the therapeutic alliance and in treatment interventions. Therapy animals are chosen specifically for their behavior, having received training necessary for them to be safe and for humans around them to be safe within the therapy session. Further, therapy animals are chosen for particular characteristics in their physical form and/or personality in order to further the therapeutic process with the (human) client.

Even with such carefully considered choices, there may be risks to clients participating in AAIs. Possible risks to clients receiving AAIs include:

Personal injury. The behavioral screening which takes place before an AAI session includes the provider’s identification of the animal’s interest in and consent to participate with each client. Further, it is the provider’s responsibility to observe the therapy animal’s behavior during the session to avoid undue stress on the animal which can lead to possibly injurious behavior. Nonetheless, injury may occur. An injury may be as obvious as a scratch or a bite, which are largely preventable with the correct level of the provider’s attention to the therapy animal’s needs. An injury may also be as unexpected (and unpreventable) as a client biting her own tongue while hugging an animal who suddenly sits up or pulls out of the hug.

Zoonotic infection or disease. A zoonotic disease is a disease a human may acquire through contact with an animal. (Diseases animals may acquire through contact with humans are called reverse zoonoses). It is the provider’s responsibility to assess the therapy animal’s health prior to asking the animal to work, and this includes a physical examination as well as a behavioral assessment that aligns with recommendations from reputable sources such as those provided by Murthy et al., (2015) [45] and Howie (2015) [46]. This assessment is conducted prior to each work period and on an ongoing basis. As a result, the risk of zoonotic transmission is greatly reduced when the provider is trained in what to observe physically and behaviorally as indicators of illness or disease.

Allergies. It is the mental health provider’s responsibility to inquire about possible client allergies to the species of therapy animal being worked with prior to beginning an AAI session. The provider is dependent upon the accuracy of the client’s knowledge and response, which means that a provider could be surprised by a client’s allergic response. Conversely, a client should never be surprised by the presence of a therapy animal in the provider’s office.

Fears or phobias. Similarly, prior to beginning an AAI session it is the mental health provider’s responsibility to inquire about possible animal fears or phobias held by the client. This common risk-management procedure greatly reduces the risk of a client phobia to the species of therapy animal.

Traumatic memories. Animal abuse may be part of a client’s history, whether the client perpetrated the abuse personally or was forced to observe the abuse. It is possible that interacting with a therapy animal or even an animal that vaguely resembles an animal from one’s past may bring up previously unexplored themes. The emotional and physical safety of both the client and animal are paramount and a therapy animal may need to be removed from a session as a result of an unforeseen trauma response if it is not safe nor therapeutic to work through the response in the moment.

Grief. A therapy animal may need to stop work unexpectedly as a result of illness, death, or a change in the handler’s ability to participate. The provider must be prepared to deal with a client’s grief response to the ending of this relationship.

In addition, other ruptures to the clinical or therapeutic relationship may occur despite a robust informed consent process. The animal or handler may be found liable for harm occurring during treatment. The client may no longer be appropriate for AAI, notwithstanding the provider’s endorsement of the client’s appropriateness for AAI. The therapy animal may no longer be suitable or able to provide AAI. The very variability and real-world interactions that can make AAI a rich therapeutic environment may also lead to challenges in the therapeutic relationship.

### 6.2. Ameliorating Potential Risks to the Therapy Animal

Prior to endorsing the benefits of AAIs, the provider must ascertain that the therapy animal is healthy and compliant with veterinary care including vaccinations necessary to prevent intraspecies and zoonotic diseases, assure freedom from parasites, and is up-to-date with any necessary medications. Potential risks to therapy animals include:

Injury from interactions in the therapy setting. It is incumbent upon the provider to provide AAI only to clients who are appropriate to interact with an animal. Clients who are psychotic or hallucinating are contraindicated for AAI because of their significant potential to injure an animal when they are disoriented and should not be included in AAI work. Further, a therapy animal may be injured intentionally by a client who wants to hurt the provider (or handler) through the therapy animal. In contrast, a therapy animal may be hurt unintentionally by being hit by a door when a human comes through quickly or stepped on by humans (clients or colleagues) who are not accustomed to having to watch where they place their feet.

Stress. It is essential to acknowledge that AAI work can be stressful to the therapy animal [47]. Working is different from being at home with familiar people. Prolonged stress can lead to illness in therapy animals just as it can in humans. Breaks and constant assessment of stress level should be maintained at all times.

Undue stress. It is important to emphasize that therapy animals are at risk of undue stress from potential extended exposure to difficult clients or stressful environments. The therapy animal does not have the same techniques available to her as the provider does for debriefing and letting go of the pressures of the work. As a result, both the provider and handler (if they are separate people) are responsible to recognize and respond to the therapy animal’s stress behaviors and to provide methods to remediate that stress before, during, and after each work period [47,48].

AAI work does not have to cause distress, and it is the provider’s responsibility to take steps to manage and minimize stress to the therapy animal as stress can impact an animal’s health and well-being. Specifically, stress can increase the incidence of illness, and chronic stress in some cases can lead to death [49]. If the handler is separate from the mental health provider, both share responsibility. Stress-management techniques include but are not limited to:

Obtaining consent prior to the AAI session. The therapy animal must give her consent to work. If the therapy animal does not give consent (or appears ill or injured), it is the provider’s responsibility to give the animal the day off.

Performing a behavioral assessment prior to the work period. The provider and/or handler must know what behaviors in the species and specific therapy animal being worked with indicate willingness to participate. Each animal, even of the same species, can express themselves through different behaviors. Knowing one’s animal and the signs of stress they exhibit such as whining, leaning away, or panting, are important to assess.

Performing a nose-to-tail physical examination to look for signs of illness, injury, or parasites prior to the work period.

Providing an on-going healthy lifestyle that includes healthful food, adequate physical and mental exercise, activities the animal enjoys, and appropriate veterinary care.

Providing a safe and quiet client-free zone in the workplace for rest that is always freely accessible to the therapy animal.

Additional strategies and considerations are found in the Therapy Animal’s Bill of Rights [46] and within the APA-HAI Competencies document [50].

### 6.3. Ameliorating Risks to the Psychologist

While the psychologist’s code of ethics admonishes that “In those emerging areas in which generally recognized standards for preparatory training do not yet exist, psychologists nevertheless take reasonable steps to ensure the competence of their work…” (APA Code of Ethics 2.01 Boundaries of Competence), “there remains a dearth of clinically sound, rigorous, and relevant research such as that based on systematic review, with reasonable effect sizes, and statistical significance” [51]. Consequently, providers could be found liable for practicing outside their scope of competence, secondary to the lack of evidence-based protocols. They could face the potential need to legally defend their choice to provide AAT, explaining their decision and the ways they reached this option. Adverse client outcomes due to the use of an under-validated treatment such as AAT could rupture the clinical or therapeutic relationship, especially since the animal and its owner/handler could be found liable for any of the aforementioned (despite the fact that the provider has endorsed the client’s presumed capacity to benefit from AAIs, as well as the animal’s suitability to provide AAIs, through a thorough informed consent process). Public and collegial trust in the practitioner’s competency to practice within their scope of expertise may be eroded. The clinician would also be held liable for any injury or illness caused by the animal to the client. Should any of these occur, the provider’s future liability insurance could be difficult to obtain, or increase in cost, and there could be sanctions taken by various insurance providers.

An AAI provider may experience some unusual risks not typically found in other therapeutic approaches. For example:

The beloved conundrum. When providers work with their own animal as a therapy animal, this animal is often perceived as beloved rather than with the professional distance of a professional colleague. This can lead to a bias in favor of the animal, feelings being hurt if a client does not like the animal, or inability to continue to work with a client who injures the animal.

Possibly conflicting scheduling priorities. When working with a therapy animal, the animal will need breaks for exercise and elimination that might not be at the same time as the breaks the provider needs. If the therapy animal becomes ill or injured, the provider will need to leave work to attend to the animal’s needs.

Adjusted professional identity. When a provider loses a therapy animal, whether through death or a change of career on the animal’s part, the provider may feel a sense of imbalance without the therapeutic support and presence of the animal in sessions. If the provider has focused their practice on AAIs, the loss of that approach may leave them feeling adrift as they process the grief and change of identity.

Square footage. Depending on the size and species of the therapy animal, the provider may need additional square footage in the therapy room to be able to accommodate the animal’s needs, both for therapeutic interventions and for client-free rest times.

### 6.4. Ameliorating Supervisor Risk

Supervising students training in AAIs in practice or internship placements, or while accruing hours for licensure has formerly lacked consistent standards and guidelines. While not specifically developed for supervisors, the APA-HAI Competencies document serves as a means of developing a common language and standards whereby supervisors and trainees can discuss growth and goals for clinical development. It also creates a go-to document for trainees to conduct self-assessments of skills, knowledge, and attitudes.

### 6.5. Ameliorating Academic/Programmatic Risk

Up to this point, academic programs in AAI and/or that teach AAI have run the gamut from rigorous and well-designed, to poor at best. Having had no standard language and list of guidelines from which to base curriculum, there exists great diversity in what is taught and how it is taught. A successful and well-designed academic program will have measurable goals and learning objectives in place so that student outcomes and learning can be directly measured. The APA-HAI document creates an aspirational document of competencies and guidelines from which academic programs and institutions can develop curricula and measurable objectives. It also creates a framework for how we might conceptualize the relational impact and welfare of nonhuman animal partners in the work. The risks described here are a thorough but not necessarily exhaustive list.

## 7. Conclusions

AAI is a quickly growing and evolving area of practice within counseling psychology. While growth in both academic and clinical settings has led to an increased awareness of the field and professionalization, common language, standards of practice, competencies, and clear guidelines for animal welfare have lagged behind. The competencies described within this article were developed by fourteen AAI experts in collaboration with the APA Division 17, Section 13 Human-Animal Interaction Competencies Workgroup. This article, in conjunction with those recommended competencies, offers a comprehensive response to these needs. Based upon and aligned with existing APA Ethics Code 2.0, these competencies are useful and appropriate for use by practitioners and other mental health providers intending to integrate animals into their practice. Consisting of knowledge, skills, and attitudes, they are considered aspirational in nature, and not standards or requirements, per se. In addition to clinical settings, these competencies can be useful in guiding students-in-training, supervisors, organizations and administrators intending to integrate animals into their settings, as well as academic programs.

## Figures and Tables

**Figure 1 vetsci-08-00303-f001:**
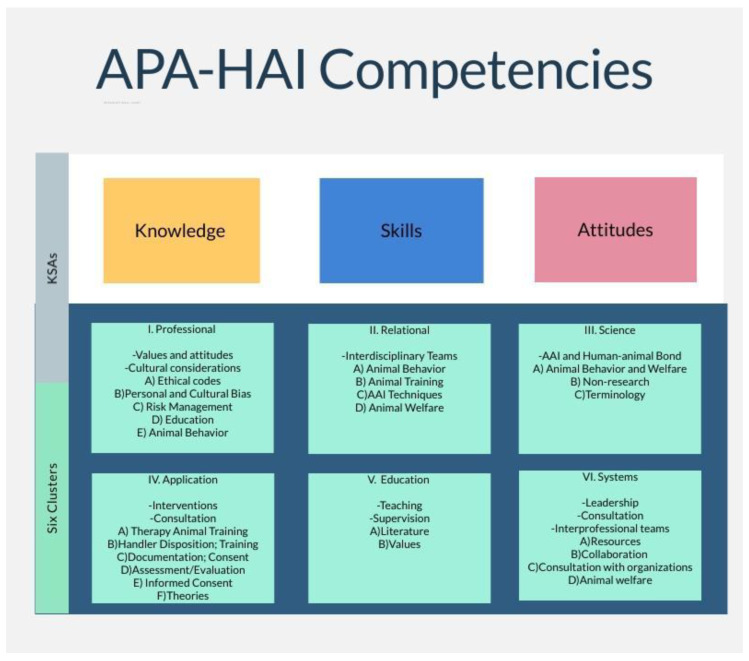
APA-HAI competencies: knowledge, skills, and attitudes.

## Data Availability

The study did not report any data.

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
