# Peer review of "Recommendations for Transdisciplinary Professional Competencies and Ethics for Animal-Assisted Therapies and Interventions"

_vetsci, 2021, doi:10.3390/vetsci8120303_

Round 1
Reviewer 1 Report
This is a well-written and important paper. I have a few comments/questions for minor improvement.
Line 12 - awkward, reword.
Line 25 + - I was not clear in the opening why the focus was on psychology, and a paragraph down, it opens broader. If the underlying intent is psychology then maybe this needs to go in the title of the paper. Though I am not sure it is.
There are other studies that looked at what social workers, for example, know about AAIs (e.g., Chalmers, Rigley-Curtiss, etc.)
I agree with the link to One Welfare, though the linkage to One Health may be oversimplifying it. I wonder though is attention here should also be given the 5 domains (and previous 5 freedoms) as well.
I am not sure why AAT / AAA is not defined. The front section reads as though we are looking at the incorporation of mainly dogs in psychologists' and others work, but then it speaks later about handlers and dogs in a session. I think it needs to be made clear to the reader up front what types of activities are included here (it is much broader than what the reader is originally led to believe).
Line 474 - Can you add a source(s) that speaks to stress leading to illness in animals, specifically therapy dogs? Other points could be better substantiated with literature too.
Line 488 - Completely agree that the animal needs to give consent, but what does this mean/look like for the uneducated reader (which many readers of this article may be). I think the entire paper can be read with this comment in mind.
I wonder if this is a working document, not the end all be all 'answer'; might want to frame it more like this. I also wonder if sending feedback to the authors as individuals review and put these competencies into practice could be a suggestion. again, so this is a living document.
Author Response
Dear Hadley Feng, Veterinary Sciences Editorial Team, and Reviewers:
Thank you for the opportunity to revise our paper and resubmit. We have carefully read and integrated your feedback and outlined below how we have done so. We appreciate the attention to our project.
Response to Feedback from Reviewer 1:
Thank you for your positive comments regarding the usefulness of our paper. We have read and incorporated your feedback as follows:
-We have reworked each sentence noted as awkward or where an edit was requested.
-We addressed the applicability to Psychology but also the generalizability to other professions to clarify our intended audience.
-We have added further background regarding the five freedoms to the One Welfare section.
-We have further clarified key terms and made it clear to which activities these recommendations apply.
-We have added many additional outside sources of support throughout the paper to include:
Lines 25-39: https://www.apa.org/ethics/code
Serpell, J.A. Animal-assisted interventions in historical perspective. In Handbook on Animal-Assisted Therapy.
Theoretical Foundations and Guidelines for Practice, 5th ed.; Fine, A.H., Ed.; Academic Press: Cambridge,
MA, USA, 2019.
Serpell, J., McCune, S., Gee, N., & Griffin, J. A. (2017). Current challenges to research on animal-assisted interventions. Applied Developmental Science, 21(3), 223-233.
Fine, A. H., Beck, A. M., & Ng, Z. (2019). The state of animal-assisted interventions: addressing the contemporary issues that will shape the future. International journal of environmental research and public health, 16(20), 3997.
Hooker, S.D.; Freeman, L.H.; Stewart, P. Pet therapy research: A historical review. Holist. Nurs. Pract. 2002,
17, 17–23.
Parish-Plass, N. Order out of chaos revised: A call for clear and agreed-upon definitions differentiating
between animal-assisted interventions. Retrieved April 2014.
Lines 59-60:
Fine, A. H., Beck, A. M., & Ng, Z. (2019). The state of animal-assisted interventions: addressing the contemporary issues that will shape the future. International journal of environmental research and public health, 16(20), 3997.
Serpell, J., McCune, S., Gee, N., & Griffin, J. A. (2017). Current challenges to research on animal-assisted interventions. Applied Developmental Science, 21(3), 223-233.
Lines 87-88:
Beetz, A. M. (2017). Theories and possible processes of action in animal assisted interventions. Applied developmental science, 21(2), 139-149.
Chandler, C. K. (2018). Human-animal relational theory: A guide for animal-assisted counseling. Journal of Creativity in Mental Health, 13(4), 429-444.
DeMello, M. (2021). Animal-Assisted Activities. In Animals and Society (pp. 233-254). Columbia University Press.
Lines 147-155:
Kogan, L., Johnson, A., Miller, C., Kieson, E., Wycoff, K., & Holman, E. (2019). Animal-Assisted Interventions: Competencies and Ethics. Proceedings of the American Psychological Association, Chicago, IL, USA, 8-11.
Fine, A. H., Beck, A. M., & Ng, Z. (2019). The state of animal-assisted interventions: addressing the contemporary issues that will shape the future. International journal of environmental research and public health, 16(20), 3997.
Trevathan-Minnis, M., Schroeder, K., & Eccles, E. (2021). Changing with the times: A qualitative content analysis of perceptions toward the study and practice of human–animal interactions. The Humanistic Psychologist.
Serpell, J.A. Animal-assisted interventions in historical perspective. In Handbook on Animal-Assisted Therapy.
Theoretical Foundations and Guidelines for Practice, 5th ed.; Fine, A.H., Ed.; Academic Press: Cambridge,
MA, USA, 2019.
Lines 237*
I am not exactly sure what type of citations we would need here. There is no main legislating body is the issue...do we need to "prove" that (cite that?) Where?
Lines 207*
Hartwig, E. K., & Smelser, Q. K. (2018). Practitioner perspectives on animal-assisted counseling. Journal of Mental Health Counseling, 40(1), 43-57.
Stewart, L. A., Chang, C. Y., & Rice, R. (2013). Emergent theory and model of practice in animal-assisted therapy in counseling. Journal of Creativity in Mental Health, 8(4), 329-348.
Respectfully,
Dr. Melissa Trevathan-Minnis
Reviewer 2 Report
Dear authors,
Thanks for your interesting manuscript. It has a depth of content and therefore does not read like a novel (but that is not necessary either). I have no comments on the content, but the scientific nature of the manuscript could be improved by including more references. In some parts, paragraphs, those references are missing. References should be added in subsequent sections: lines 25-39; lines 59-60; Lines 87-88; lines 147-155; lines 201-224; lines 231-241; etc.
A few smaller suggestions:
Line 25-57: In the introduction, it is not clear which animals (species) are focused on (horses, dogs, other pets?). It would be good to describe this in a few sentences in the introduction.
Line 68: delete the initials.
Line 99: delete 'p. 84'
Lines 526, 531, 535 and 540: the first words of each paragraph contains subtitles?
1. What is the main question addressed by the research? AAI has grown in the last decade, but more consistent language and standards are needed.
2. Do you consider the topic original or relevant in the field, and if so, why? AAI is a transdisciplinary field that has grown extremely. The manuscript describes the history, gaps, limitations and recommandations of AAI. 3. What does it add to the subject area compared with other published material? The competencies described within this article were developed by fourteen AAI experts in collaboration with the APA Division 17, Section 13 Human-Animal Interaction Competencies
Workgroup. The content of the manuscript is supported by a large group of experts. This gives a strong added value to the manuscript.
4. What specific improvements could the authors consider regarding the methodology? The methodology/structure of the manuscript is OK. First a brief history of AAI is described, then the gaps and limititations and finally the recommandations of competencies and guidelines.
5. Are the conclusions consistent with the evidence and arguments presented and do they address the main question posed? The conclusions are well described and clear.
6. Are the references appropriate? The references are appropriate, but in some paragraphs references are missing and must be added.
7. Please include any additional comments on the tables and figures. Figure 1 is OK, but more text/description/explanation in the caption of Figure 1 is needed.
Author Response
Dear Hadley Feng, Veterinary Sciences Editorial Team, and Reviewers:
Thank you for the opportunity to revise our paper and resubmit. We have carefully read and integrated your feedback and outlined below how we have done so. We appreciate the attention to our project.
Response to Feedback from Reviewer 2:
Thank you for your feedback and suggestions.
We’ve added your changes as requested as well as many additional outside sources of support as outlined above. We appreciate your suggestions.
Respectfully,
Dr. Melissa Trevathan-Minnis
Reviewer 3 Report
A good contribution to the field.
Author Response
Dear Hadley Feng, Veterinary Sciences Editorial Team, and Reviewers:
Thank you for the opportunity to revise our paper and resubmit. We have carefully read and integrated your feedback and outlined below how we have done so. We appreciate the attention to our project.
Reviewer 3:
Thank you for your confidence in our paper. We appreciate your review.
Please let us know if there is anything further needed.
Respectfully,
Dr. Melissa Trevathan-Minnis
Round 2
Reviewer 2 Report
Dear authors,
Thank you for incorporating the suggestions into the new version of the manuscript. However, I have one more comment: line 39: '...any activity or work involving animals.': I think you mean '...any activity or work involving DOMESTIC animals.' Not? Or does this also include spiders, tigers, snakes,...?
Kind regards
Author Response
Thank you for your comment and question. Yes, this is tricky and I understand your position.
While domestic animals are most typical, there are also practitioners that partner with farm animals, reptiles, and animals we might not consider trainable. We'd suggest working with less "common" animals with care and intention, but would still argue that ethical and humane practice is critical for any animal involved in a working relationship and that the standards apply in all cases. We'd like to keep the article without the distinction of "domestic" if that is okay, but we are open to another perspective as well.
Respectfully,
Melissa Trevathan-Minnis